# Research on Flexural Bearing Capacity of Reinforced Hollow Slab Beams Based on Polyurethane Composite Material Positive and Negative Pouring Method

Jin Li [1],*, Tiancheng Zhou [1], Xiang Li [2],*, Dalu Xiong [3], De Chang [3], Zhongmei Lu [3] and Guanghua Li [4]

1 School of Transportation Civil Engineering, ShanDong JiaoTong University, Jinan 250357, China
2 Shandong Hi-Speed Construction Management Group Co., Ltd., Jinan 250001, China
3 Jinan Kingyue Highway Engineering Company Limited, Jinan 250220, China
4 Shandong Huajian Engineering Testing Co., Ltd., Jinan 250132, China
* Correspondence: sdzblijin@163.com (J.L.); sdyslixiang@163.com (X.L.)

**Abstract:** In order to explore the construction technology of prestressed steel strand–polyurethane cement composites for strengthening hollow slab beams, two reinforced test beams (L1, L2) and one unreinforced test beam (L0) were subjected to flexural static load tests. The deflection, ductility, stiffness, strain, and bearing capacity of each test beam were used to summarize the influence of different reinforcement techniques on the flexural performance of hollow slab beams. Research shows the prestressed steel strand–polyurethane composite material was well-bonded to the hollow slab beam, which effectively inhibits the development of concrete cracks and delays the damage process of hollow slab beams, that the reinforcement effect of the test beam L1 under the reverse pouring process was remarkable, and the bending performance of the test beam L2 under the forward pouring process of the simulated real bridge was good, which was much better than that of the unreinforced beam L0. The use of tensile prestressed steel strands and forward casting of polyurethane–cement composite materials effectively improved the flexural bearing capacity of the test beams, and this reinforcement process can be further extended to engineering applications.

**Keywords:** steel strand polyurethane cement; hollow slab beams; reinforcement process; bending test; bearing capacity





## 1. Introduction

With the increase in road traffic and vehicle load, there has been an increase in damage to older bridges, which seriously affects the safety of traffic travel and makes the field of bridge reinforcement more and more important. Reinforcing older bridges can not only improve the bearing capacity of a bridge, but can also ensure transportation efficiency, and is in line with the green and sustainable development strategy.

Many scholars are carrying out research on bridge reinforcement: Tang Yu [1] proposed a combined reinforcement technology of embedded CFRP bars and external prestressed steel wire ropes to reinforce concrete beams, which effectively improved the cracking load, yield load, ultimate load, and bending stiffness of concrete beams without reducing ductility; Guo Yanru [2] studied the reinforcement method of cracked beams and slabs by externally prestressing unbonded steel strands, and proved the practicality of this process to narrow cracks and improved the flexural bearing performance of concrete beams; Zhang Yang [3] used the reinforcement technology of ultra-high performance concrete prestressed UHPC to reinforce RC, and reported that the prestressed UHPC reinforcement layer could improve the structural stress state and inhibit the formation and development of cracks; the team of Deng Langni [4] conducted bending static load tests on carbon fiber plate reinforced beams with embedded fiber grating strain sensors and unreinforced comparative beams, and verified that this process could effectively delay the fatigue

fracture time of steel bars and improve the availability of the fatigue life of the test beams; Zhang Shengran [5] studied the reinforcement mechanism and advantages of polyurethane cement reinforcement materials and steel wire ropes on T beams, compared the results of the flexural performance tests of each beam to analyze the reinforcement effect, and proved the practical value of polyurethane cement and steel wire ropes to strengthen T beams. Zhengpeng Yang [6] prepared a flexible and stretchable polyurethane/water glass grouting material. The grouting material showed excellent thermal stability and crack repair performance, and provided a basis for the ratio and process of polyurethane materials for repairing cracks.

Although the above research can effectively improve the bearing capacity of the bridge and prolong the bridge service life, it also has the disadvantages of increasing the self-weight of the bridge, difficult construction, and high economic cost. In the field of polyurethane cement composite materials combined with prestressed steel strands to strengthen hollow slab girders, there is a lack of reliable test experience and test data based on reliable models and real bridge load tests, and no systematic research has been made to verify the effectiveness of this reinforcement method.

In recent years, polyurethane–cement composites have gradually been applied in the field of bridge reinforcement, and the combination of prestressed steel to strengthen bridge superstructures is a hot research topic. Based on this, this paper used the combination of prestressed steel strands and polyurethane cement composites to strengthen hollow slab beams and explored the toughening construction technology and toughening effect of polyurethane cement composite materials on hollow slab girders under engineering simulation conditions to promote the subsequent promotion of prestressed steel strand. The combination of polyurethane cement composite materials to strengthen the hollow slab solid bridge provides theoretical and data reference, has high economic value and practical value, and provides a basis for engineering practice in the field of hollow slab solid bridge reinforcement.

## 2. Trial Overview

### 2.1. Experiment Material

(1)　Concrete

The concrete grade was C50, the mix ratio was water:cement:sand:stone of 0.42:1:1.152:2.449, respectively; the average compressive strength of the 150 mm × 150 mm × 150 mm cube specimen under this mix ratio was 52.6 MPa.

(2)　Rebar

Four longitudinal steel bars were arranged in the compression zone and tension zone of the hollow slab beam, and one stirrup was arranged at every 15 cm interval along the longitudinal steel bars. Two support ribs were arranged at the supports at both ends. The specific material parameters of the steel bar are shown in Table 1.

**Table 1.** Rebar material parameters.

| Rebar Category | Rebar Model | Diameter (mm) | Yield Strength (MPa) | Design Value of Tensile Strength (MPa) | Elastic Modulus (MPa) | Poisson's Ratio |
|---|---|---|---|---|---|---|
| Longitudinal reinforcement | HRB400 | 16 | 400 | 330 | $2 \times 10^5$ | 0.3 |
| Stirrups | HRB335 | 12 | 335 | 280 | $2 \times 10^5$ | 0.3 |
| Stiffener | HRB400 | 16 | 400 | 330 | $2 \times 10^5$ | 0.3 |

(3)　Polyurethane cement composite

The polyurethane cement composite materials were composed of isocyanate (content ≥99%, density 1.23 g/cm³), modified polyether (HYPOP-3628, hydroxyl value 25–29 mgKOH/g, moisture ≤0.08%, viscosity (25 °C) 2200–3000 mPa·s), cement, defoamer,

catalyst, and other components [7–12]; the cement adopted composite Portland cement with the grade of 42.5; the defoamer formula was modified by the manufacturer, which reduced the generation of bubbles during the reaction process; the catalyst was prepared by the manufacturer and effectively controlled the chemical reaction rate of isocyanate and modified polyether. The proportion of the polyurethane cement composite material is shown in Table 2, and the density of the formed test block was 1400 kg/m$^3$.

**Table 2.** The mass ratio of each component.

| Composition | Proportion (%) |
| --- | --- |
| Isocyanate | 30.5 |
| Modified polyether | 35 |
| Cement | 33.2 |
| Defoamer | 0.5 |
| Catalyst | 0.8 |

(4)  Prestressed steel strand

The steel strands used in the toughening of the hollow slab beams [13,14] are shown in Table 3 for the type, size, and mechanical parameters.

**Table 3.** Specifications and parameters of the steel strands.

| Category | Nominal Diameter (mm) | Effective Cross Section (mm$^2$) | Design Value of Tensile Strength (MPa) | Elastic Modulus (GPa) | Poisson's Ratio |
| --- | --- | --- | --- | --- | --- |
| 1 × 7 Standard | 15.2 | 140 | 1860 | 0.195 | 0.3 |

### 2.2. Test Beam Design

A total of three test beams numbered L0~L2 were made. The dimensions of each test beam were the length, width, and height of 300 cm, 50 cm, and 40 cm, respectively. The cross-section of the hollow plate beam was arranged with a circular channel with a diameter of 25 cm; supports were cast at both ends of the hollow slab beam. The length, width and height of the supports were 50 cm, 15 cm, and 10 cm, respectively. Tension holes for the interspersed steel strands were reserved at the L0~L2 supports of the test beam. The specific dimensions of the test beam are shown in Figures 1 and 2.

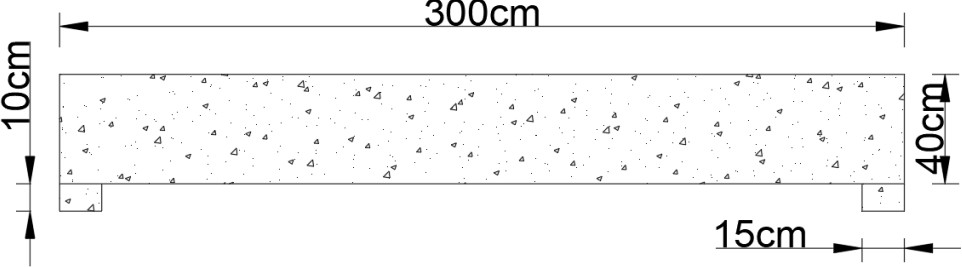

**Figure 1.** Longitudinal section of the test beam.

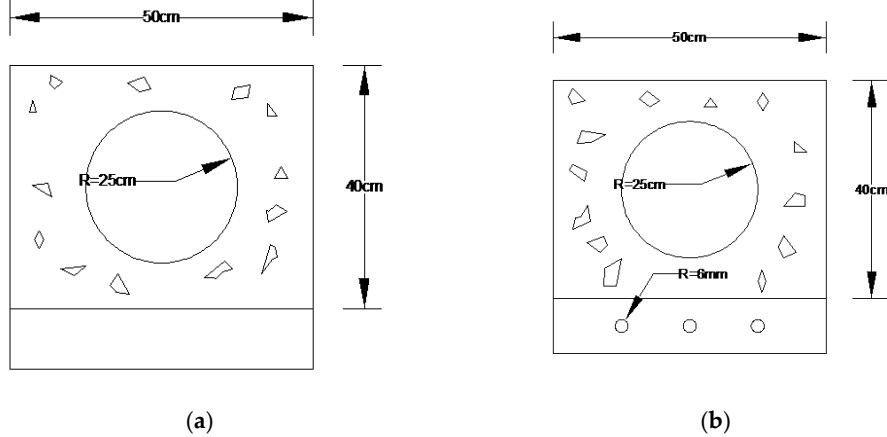

(**a**)                                    (**b**)

**Figure 2.** Cross section of the test beam. (**a**) Cross section of test beam L0. (**b**) Cross-section of test beams L1 and L2.

### 2.3. Reinforcement Scheme

Under the premise of the same reinforcement ratio and tensile stress of steel strands, the toughening construction technology and toughening effect of polyurethane–cement composites on hollow slab beams under the simulated real bridge reinforcement state were explored. Three steel strands with a tensile stress of 300 MPa were similarly applied to the test beams L1 and L2, and the reinforcing material and thickness were the same. We mixed the polyurethane cement composite material in proportion, and stirred it with a mixer for 15 min to ensure that it was evenly mixed. Heat will be generated when the composite material reacts inside, the viscosity will gradually increase, and the surface will become skinned and smooth. At the same time, the reaction temperature was monitored and the ambient temperature was selected between 10 °C and 25 °C. The test beam L2 was used to simulate the real bridge, which was different to the test beam L1 that was poured. The toughened material pouring formwork was erected at the bottom of the beam, and the pouring thickness of the polyurethane cement composite material was uniformly controlled at 4 cm. The material was poured, and after 3 days, the formwork was removed; the flexural bearing capacity test of the hollow slab beam was carried out it after it had been maintained for 14 days. The failure mechanism and bearing performance were compared with the L0 and L1 test beams. The reinforcement scheme of each test beam is shown in Table 4.

**Table 4.** Test beam reinforcement scheme.

| Test Beam Number | Number of Strands | Tensile Stress (MPa) | Reinforcement Material | Pouring Method | Material Thickness (cm) |
|---|---|---|---|---|---|
| L0 | 0 | 0 | / | Inverted beam bottom reverse pouring | / |
| L1 | 3 | 300 | Steel strand, polyurethane cement composite | Inverted beam bottom reverse pouring | 4 |
| L2 | 3 | 300 | Steel strand, polyurethane cement composite | Simulate the forward pouring of a real bridge | 4 |

### 2.4. Measuring Point Layout and Loading Scheme

(1)　Strain measuring point

In order to better reflect the strain situation along the beam height in the midspan of the test beam under various loads, strain measuring points were arranged on one side of the midspan of the test beam. The interval between two adjacent measuring points was 6.6 cm, a total of six measuring points were arranged on the control beam L0, and seven strain measuring points were arranged on the test beams L1~L2 [15]. In order to observe

the tension change of the beam bottom under various loads, two strain measuring points were arranged at the bottom of the beam in the mid-span.

(2)    Deflection measuring point

In order to more clearly reflect the displacement changes of the key sections of the test beam under various loads, deflection measuring points were arranged at the central support, one quarter point and mid-span of the test beam. A total of five deflection measuring points were arranged for each test beam. The layout of the strain and deflection measuring points is shown in Figure 3.

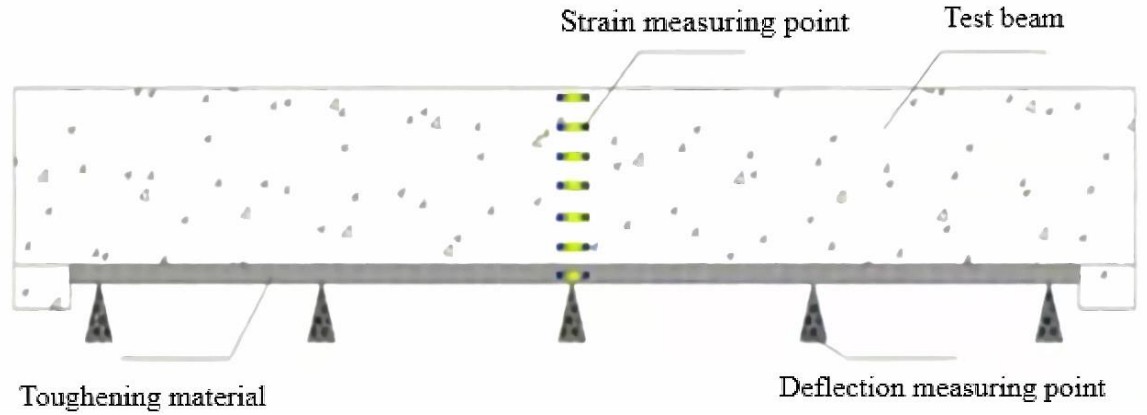

**Figure 3.** Layout of the strain and deflection measuring points.

(3)    Loading scheme

According to the layout plan of the measuring points, the resistance strain gauges were posted along the beam height, the resistance strain gauges were posted at the bottom of the beam at the mid-span position, and the deflection gauges were arranged at the position of the test beam support, the one quarter section and the mid-span position. The reaction force frame was used to load through the jack, and the load was applied to the test beam through the steel plate, pressure sensor, distribution beam, etc., and the distribution beam interval was 25 cm. Before the start of the test, an initial load of 10 kN was applied, and the static acquisition system was used to observe whether the strain gauge, displacement sensor, and pressure sensor were working normally. After the inspection was completed, the jack was unloaded, the data of the static acquisition system were cleared, and the official loading began, and the loading was 15 kN per level. During the loading process, the phenomenon of the test beam under each load was recorded, and the crack width gauge was used to observe and record the crack development of the test beam at all times, and attention was paid to the cracking sound.

### 3. Test Results and Analysis

*3.1. Destruction Phenomenon*

The three test beams were observed by the static strain acquisition instrument. Before the crack appeared, the deflection of the midspan, support. and one quarter section increased regularly with the increase in the load, and the strain value of the beam bottom increased with the load. Under the step-by-step load action of test beams L0~L2, cracks appeared, multiple cracks spread, and deflection increased, and the cracks spread to the top of the beam. The reinforcement beams L1 and L2 were also accompanied by polyurethane reinforcement when the cracks spread to the top of the beam. The phenomenon of material fracture, the polyurethane fracture at the bottom of the beam was a "h" shape. The concentrated forces corresponding to the midspan sections [16,17] of the comparative beam L0 and the reinforced beams L1~L2 in each state are shown in Table 5. The distribution of cracks in test beam L2 and the fracture at the bottom of the beam are shown in Figures 4 and 5.

**Table 5.** Loading state loads of each test beam.

| Test Beam Number | Cracks Appear | Multiple Cracks with Increasing Deflection | Fissures Grow Sharply | Toughened Material Fracture |
|---|---|---|---|---|
| L0 | 150 kN | 165 kN | 195 kN | 240 kN (crack spread beam roof) |
| L1 | 225 kN | 330 kN | 370 kN | 450 kN |
| L2 | 180 kN | 290 kN | 315 kN | 465 kN |
| Standard deviations | 30.82 kN | 70.29 kN | 73.07 kN | 102.71 kN |

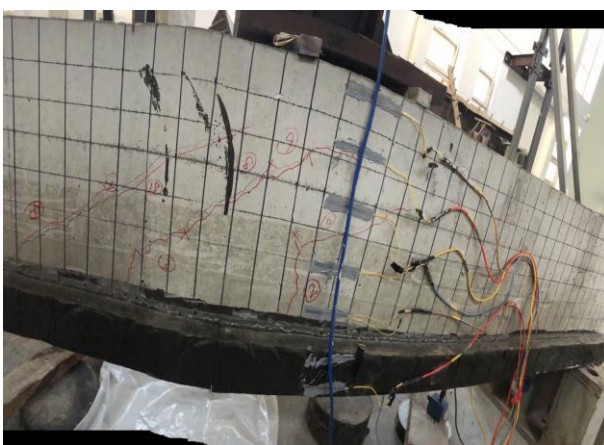

**Figure 4.** Fracture distribution.

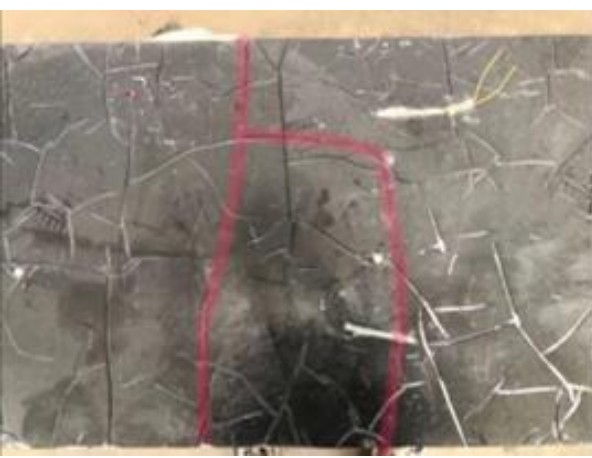

**Figure 5.** Fracture shape of the beam bottom.

*3.2. Load-Deflection*

The load–deflection curves of test beams L0~L2 are shown in Figure 6. From the figure, it can be seen that the stress failure process of the test beam was divided into three stages: a no crack stage, crack development stage and plastic development stage [18,19].

In the crack-free stage, the load–deflection curve was almost linear. The cracking loads of test beams L0, L1, and L2 were 150 kN, 225 kN, and 180 kN, respectively. Compared with test beam L0, the cracking load of test beam L2 was increased by 20%; compared with test beam L1, the cracking load of test beam L2 was reduced by 25%. Under the conditions of applying the same steel strand reinforcement ratio and steel strand tensile stress, it can be speculated that due to the reinforcement of the simulated real bridge, the forward casting method of polyurethane can be adopted. The bonding between the cement composite material and the concrete beam bottom was insufficient, so the toughening method and the concrete beam cannot be fully co-stressed.

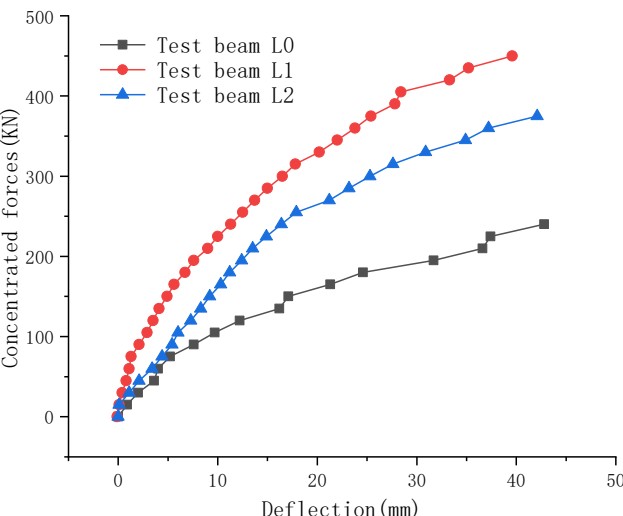

**Figure 6.** Load–deflection curve.

During the crack development stage, the slope of the load–deflection curve began to decrease gradually, showing a nonlinear trend. After cracks appeared in the test beams, the cracks continued to extend upward with the increase in load [20]. There was no damage to the polyurethane–cement composite material, nor was it peeled off from the bottom of the beam. After the tensile zone at the bottom of the beam was damaged, the prestressed steel strand and the polyurethane toughening material mainly bore the tensile stress of the bottom of the beam, constrained the deformation of the test beam, and delayed the concrete cracking. Compared with test beam L0, the yield load of test beam L2 was increased by 61.5%; compared with test beam L1, the yield load was reduced by 15.4%.

In the plastic development stage, the test beam deflection increased swiftly, the crack developed rapidly, and the crack penetrated the test beam above the neutral axis. There were two failure modes of the test beam. One was that the concrete in the compression zone above the neutral axis of test beam L0 was broken after reaching the ultimate load, and the unloading phenomenon occurred when the loading continued. The other was that after the toughened test beam L2 reached the yield load, the polyurethane–cement composite material mainly bore the beam bottom tensile stress, and the toughened material broke when the ultimate load was applied. Compared with test beam L0, the ultimate load of test beam L2 was increased by 62.5%; compared with test beam L1, the ultimate load of test beam L2 was reduced by 15.4%. It was proven that the use of prestressed steel strands and polyurethane–cement composite material at the bottom of the test beam could effectively improve the bearing capacity of the test beam, and this reinforcement method could be effectively applied in the reinforcement of real bridges. The load deflection–curve of each test beam is shown in Figure 6.

### 3.3. Bending Stiffness

The test beam was a concrete simply-supported beam, and the calculation formula of its mid-span deflection under the load effect is as follows:

$$y = \frac{5}{48} \frac{Ml^2}{EI} \tag{1}$$

Then, the formula for calculating the flexural stiffness of the test beam under load can be deduced:

$$EI = \frac{5}{48} \frac{Ml^2}{y} \tag{2}$$

Through the deflection measured during the loading process and the bending moment converted by the load, the flexural rigidity of the section of the test beam under the ultimate

load can be specifically calculated. The bending stiffness of test beam L0, test beam L1, and test beam L2 is shown in Figure 7. It can be seen from Figure 7 that the stiffness of each test beam was the smallest when it reached the yield load, and the stiffness was almost similar when it reached the cracking load and the ultimate load. This shows that as the load increased step by step, the deflection of the test beam decreased sharply when the test beam reached the yield load, and the deflection value changed rapidly [21,22]. When each test beam reached its respective cracking load, yield load, and ultimate load, compared with test beam L1, the stiffness of test beam L2 decreased by 38.3%, 59.7%, and 39.1%, respectively. On one hand, the stiffness was reduced because the toughened material was not fully bonded to the bottom of the beam. On the other hand, in order to set up the template to prevent the leakage of the toughened material, there was a 4 cm gap between the toughened section and the two ends of the test beam support, thereby reducing the gap between the test beam and the toughened material. The overall mechanical performance of the toughened material reduced the test beam stiffness. The gap between the toughened section and the support is shown in Figure 8. Compared with test beam L0, the stiffness of test beam L2 was increased by 85.3%, 47.9%, and 45.2%, respectively. It can be seen that the combined toughening method of the prestressed steel strand and the polyurethane–cement composite material greatly enhanced the rigidity of the test beam, thereby effectively improving the rigidity of the test beam and providing a higher level of stability for the subsequent application of the real bridge after reinforcement.

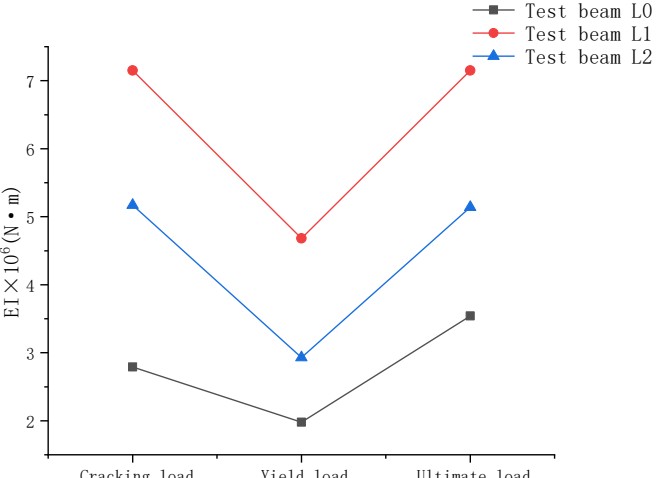

**Figure 7.** Beam stiffness in the load test at different stages.

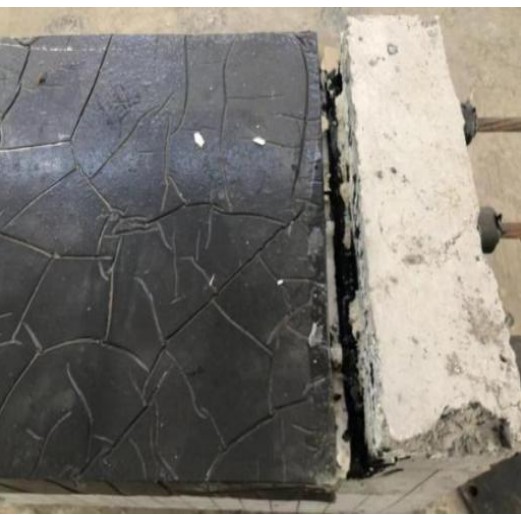

**Figure 8.** Gap between the toughened section and support.

### 3.4. Ductility Analysis

The displacement ductility coefficient μ was calculated by the ratio of the mid-span limit deflection $\delta_u$ of the test beam to the yield deflection $\delta_y$. The ductility calculation results of test beams L0, L1, and L2 are shown in Table 6, and the ductility distribution of the test beam is shown in Figure 9.

**Table 6.** Ductility parameters of each test beam.

| Test Beam | $\delta_y$/mm | $\delta_u$ | $\mu$ |
|:---:|:---:|:---:|:---:|
| L0 | 31.7 | 42.8 | 1.35 |
| L1 | 25.4 | 40.5 | 1.59 |
| L2 | 27.6 | 42.1 | 1.52 |
| Standard deviation | 2.61 | 0.96 | 0.58 |

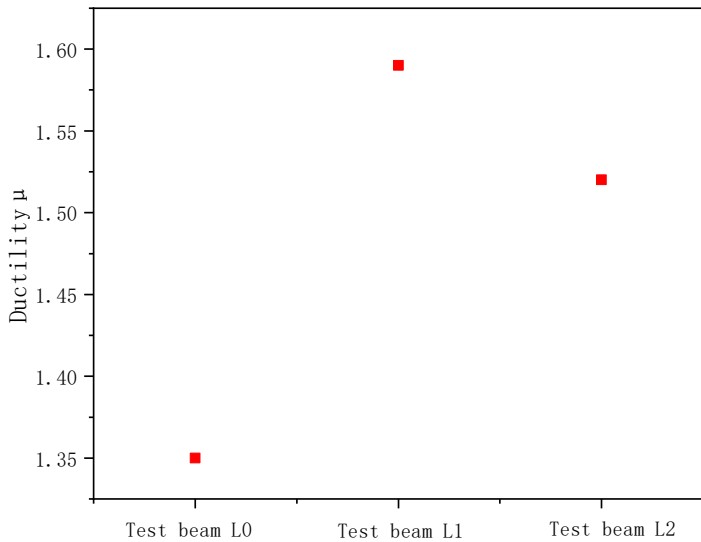

**Figure 9.** Test beam ductility.

From the ductility scatter plot, it can be seen that compared with test beam L0, the ductility of test beam L2 was increased by 12.6%, indicating that the combined toughening of the prestressed steel hinge and the polyurethane–cement composite material could effectively improve the ductility of the hollow slab beam [23] and provide a greater safety reserve for the subsequent application of real bridge reinforcement. Compared with test beam L1, the ductility of test beam L2 was reduced by 4.6%. This was because the toughening material, the prestressed steel strand, and the support did not form a good force-bearing whole, and the tensile stress of the steel strand and the polyurethane increase were purely dependent. The adhesion between the tough material and the concrete was used to delay the cracking of the test beam at the mid-span. Under the premise of applying the same reinforcement ratio and tensile stress of steel strands, the ductility of the polyurethane–cement toughened material in forward pouring was reduced, but the reduction effect was not obvious. It can be considered that, in the process of strengthening the real bridge, the combination of prestressed steel strands and forward-casting polyurethane–cement composite materials can effectively improve the ductility of the bridge and reduce the suddenness of bridge disasters.

### 3.5. High Strain along Beam

Strain gauges were posted at different beam heights on the side of the mid-span beam, and the strain values of each section under various loads were obtained through the static strain acquisition instrument, and the average distribution map of the measured strain was drawn [24,25]. It can be seen from the distribution diagram of the high strain along

the beam that when the three test beams were in the elastic stress stage, the neutral axis of the section moved upward gradually, as the load increased step-by-step. The test beam was not damaged at this time, and the strain was an inclined straight line. After the test beam L0 was cracked, the cracks gradually spread with the increase in load until the beam failed. After test beams L1 and L2 were cracked, the steel bars were peeled off from the surrounding concrete, resulting in relative displacement. The prestressed steel hinge line and the polyurethane–cement composite material gradually began to bear the main external load, offset part of the tensile stress, and restrained the deformation of the beam bottom. When L1 was damaged, the crack width in the tension zone increased sharply, the steel bars yielded within a limited length, and the concrete in the compression zone was crushed. Although test beam L2 adopted the forward pouring method, it conformed to the assumption of plane section in a similar manner to beams L0 and L1 at each stage. The distribution diagrams of the high strain along the beams of each test beam are shown in Figures 10–12.

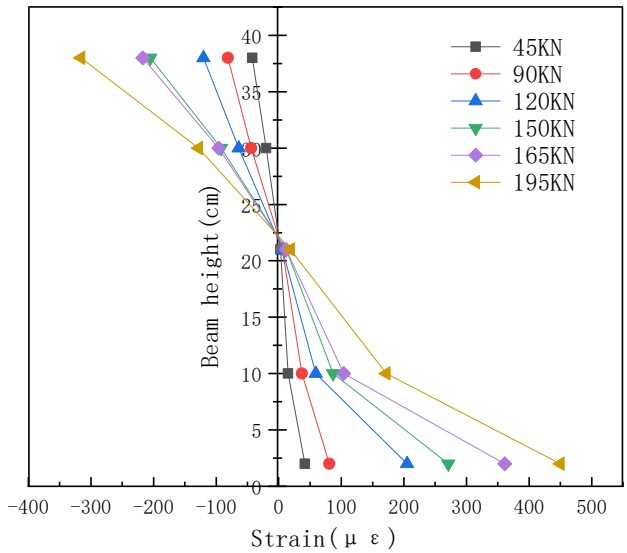

**Figure 10.** High strain along the beam (L0).

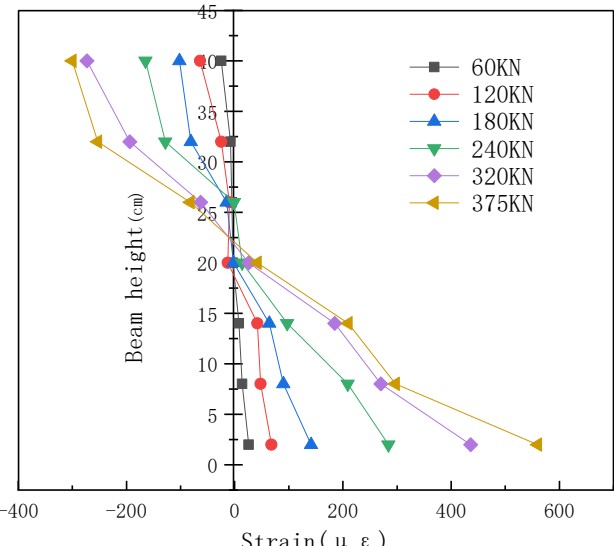

**Figure 11.** High strain along the beam (L1).

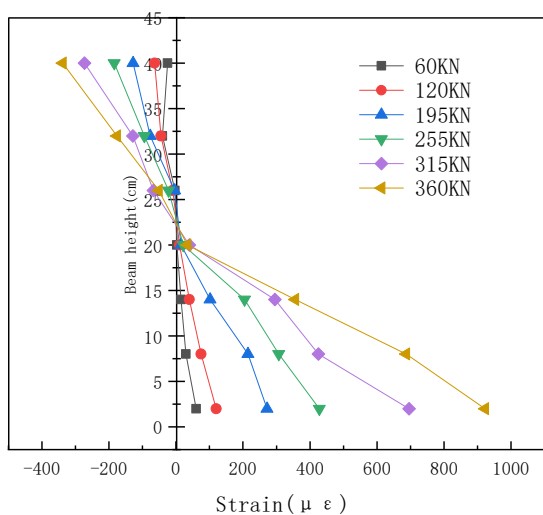

**Figure 12.** High strain along the beam (L2).

### 3.6. Microscopic Characterization Analysis

The fractured surface of the specimen after loading was scanned by a SEM electron microscope through a Zeiss sigma 500 field emission scanning electron microscope. After the fracture surface was enlarged, it could clearly show that the stress body composed of cement and polyurethane materials was in a good bonding state, and the materials were tightly connected in the microscopic environment, and the integrity was greatly improved. SEM micrographs are shown in Figures 13 and 14.

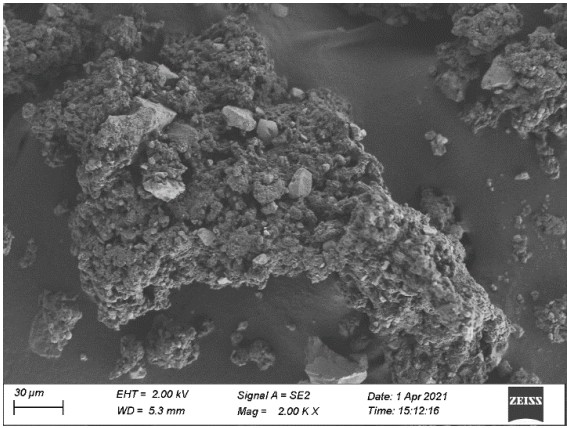

**Figure 13.** The 30 μm SEM micrograph.

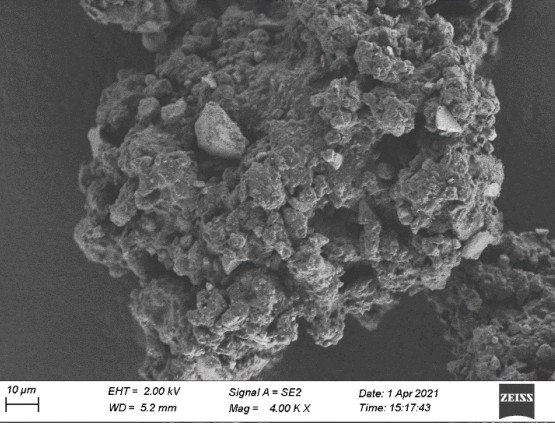

**Figure 14.** The 10 μm SEM micrograph.

## 4. Abaqus Finite Element Analysis

Using the finite element nonlinear analysis software Abaqus to simulate and analyze the three test beams, we analyzed and compared the technical indicators such as deflection and strain calculated by the simulation, studied the reinforcement effect of the test beams, and verified the accuracy and reliability of the test results. Since the failure phenomena and stress–strain distributions of the three test beams were roughly similar, test beam L2 was taken as an example for illustration.

### 4.1. Crack Distribution

The whole process of the test beam experienced stress and deformation during the loading process, which was divided into three stages, namely, the stage where no concrete cracks appeared in the test beam, the stage where cracks appeared and developed, and the stage where the test beam failed. Combined with the Abaqus model, the crack development process and distribution of the test beam under load were analyzed. The crack evolution distribution of test beam L2 is shown in Figure 15 when the load acts on the finite element model of the beam.

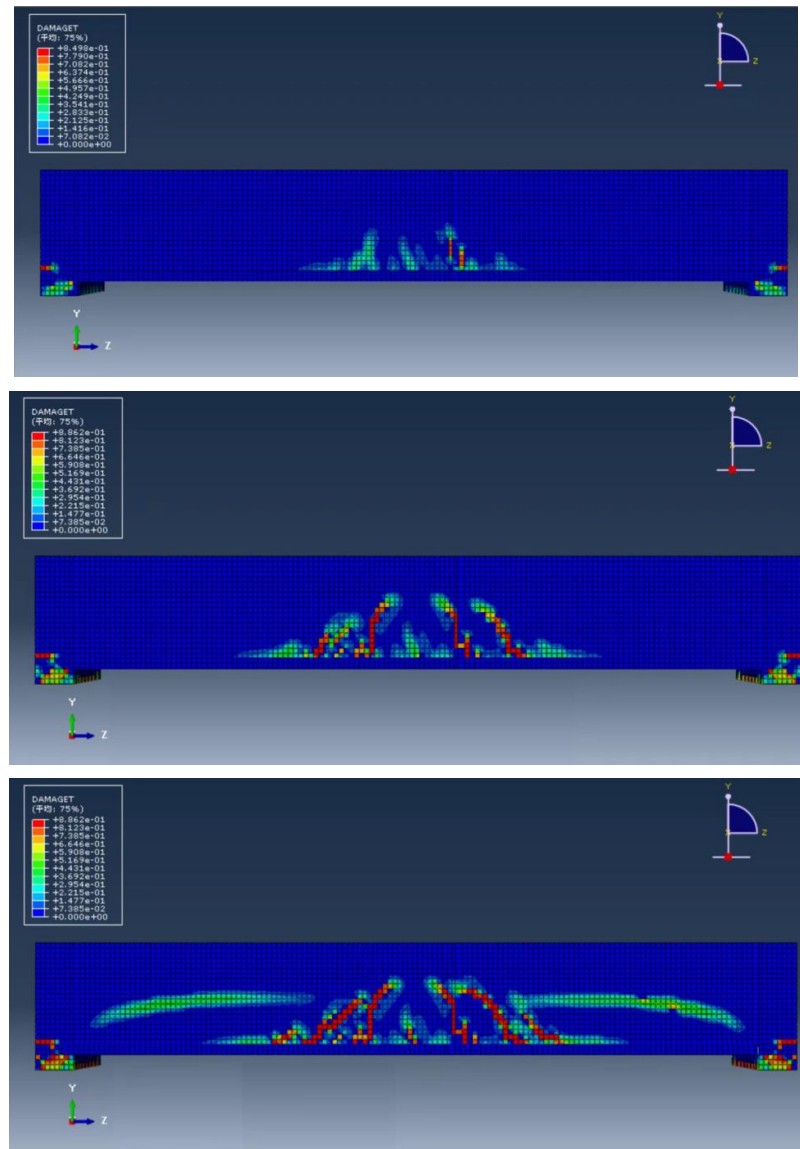

**Figure 15.** Evolution process and distribution of fractures.

### 4.2. Load-Deflection

Taking test beam L2 as an example, we adjusted the load size in the Abaqus model, respectively, by applying the cracking load, yield load, and ultimate load obtained through the test, verified the deflection produced by its mid-span section, support section, and 1/4, and tested the deflection generated by the load of the beam during the loading process, which were compared and analyzed. The deflection under finite element load simulation are shown in Figures 16–18.

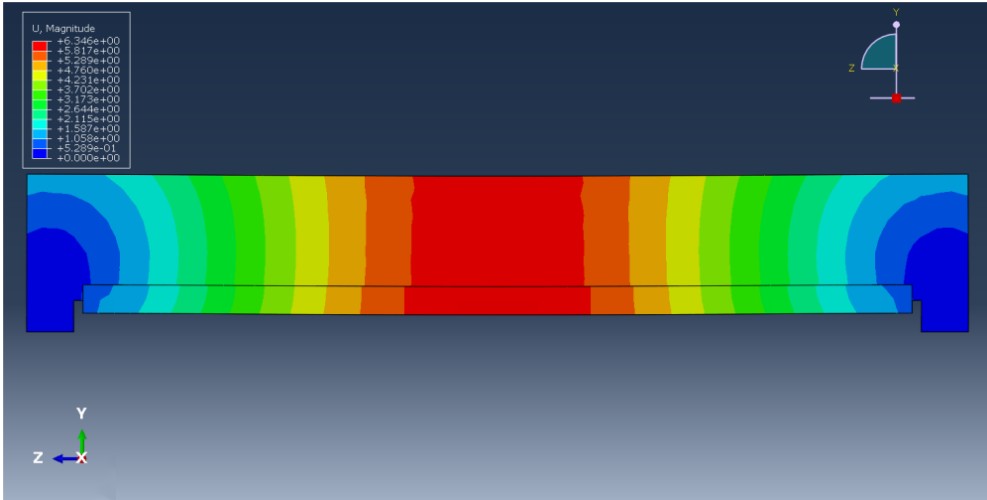

**Figure 16.** Deflection of test beam L2 when it reached the cracking load.

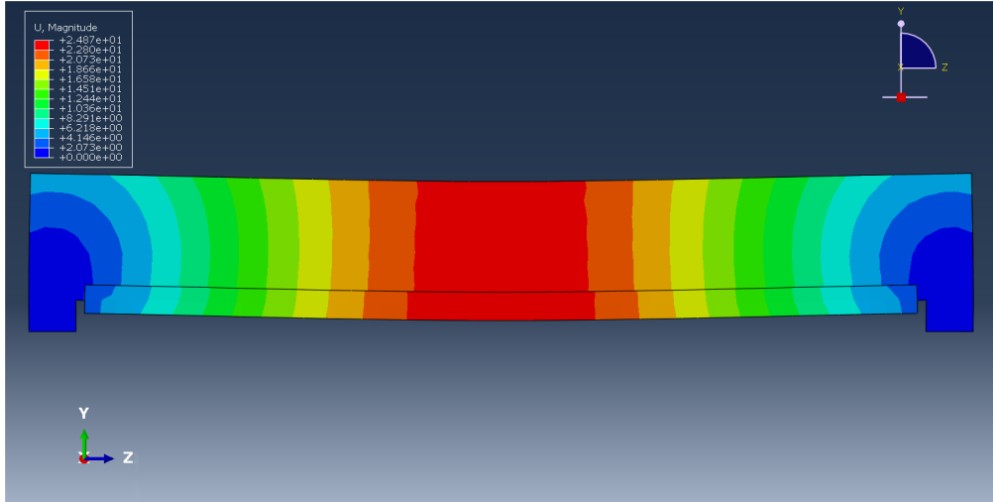

**Figure 17.** The deflection of test beam L2 when it reached the yield load.

Test beam L2 reached the cracking load, yield load, and ultimate load. The theoretical and measured deflection values corresponding to the support section, 1/4 section, and mid-span section are shown in Figure 19. The deflection theoretical value corresponding to each section was slightly different from the measured value, but it was within the allowable range of error, which shows that the deflection obtained by test beam L2 under load was reliable.

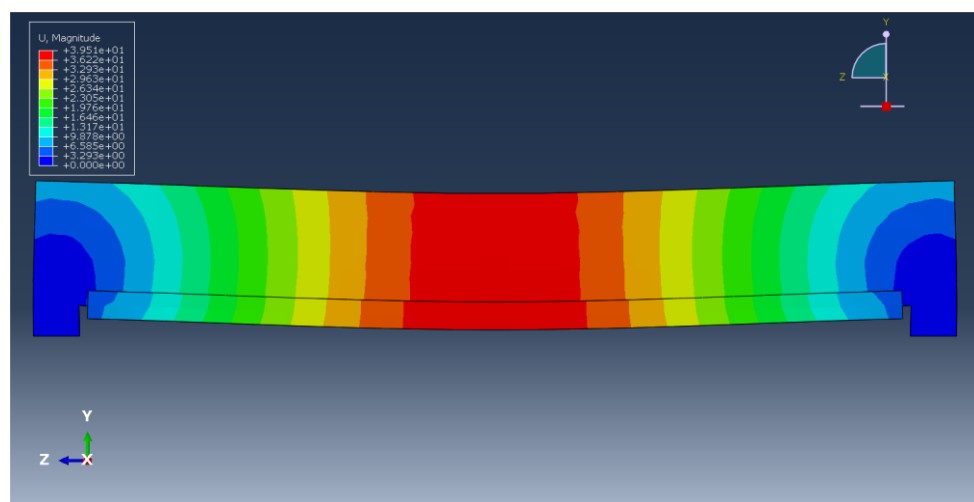

**Figure 18.** The deflection of test beam L2 when it reached the ultimate load.

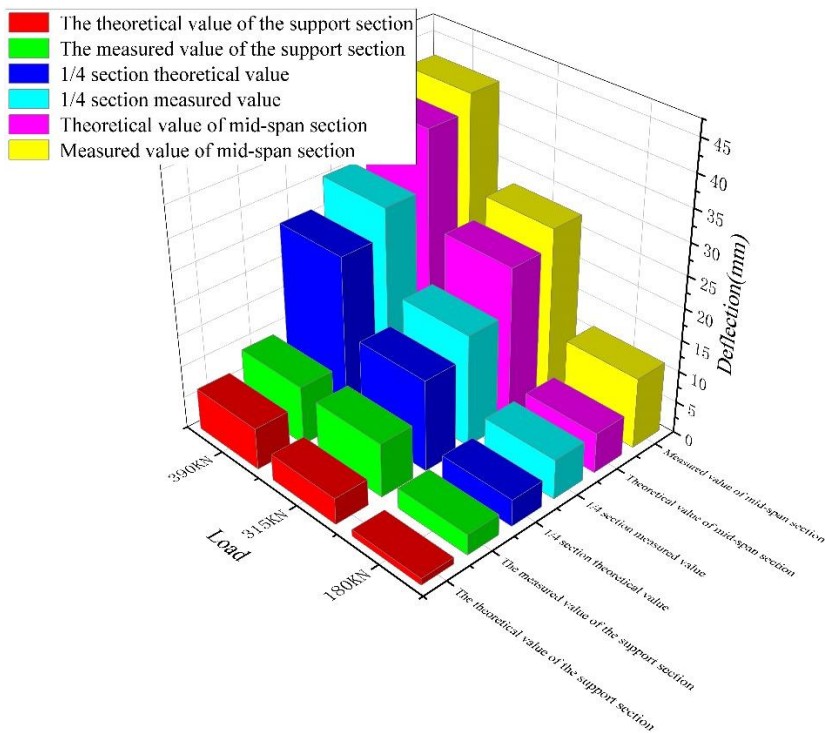

**Figure 19.** Deflection of each section of test beam L2 under special load.

The mid-span load–deflection curve of the Abaqus model of test beam L2 was analyzed. It can be seen from Figure 20 that the change trend of the theoretical value of the load–deflection curve was generally consistent with that of the measured value, and both experienced three stages of stress and deformation failure of the test beam. The change in the theoretical value simulated by finite element software could show the ideal nonlinear change, while the change of the measured value showed the contingency of the experiment, but the change trend of measured value was roughly the same as that of the theoretical value, which does not affect the explanatory nature of the test.

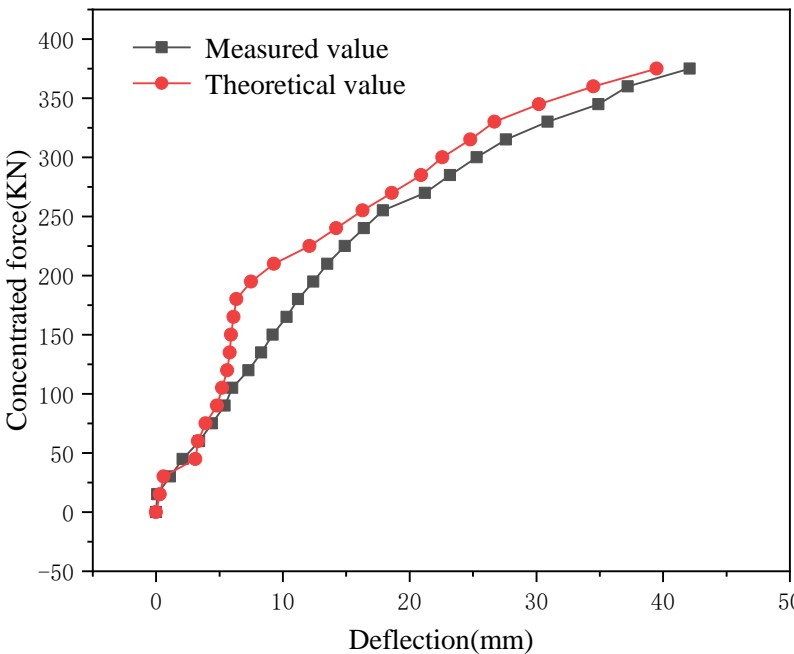

**Figure 20.** Load–deflection curves of the theoretical and measured values of test beam L2.

*4.3. Strain*

The tensile strain at the bottom of mid-span beam L2 under load was the largest, the compressive strain at the top of the beam was the largest, and the strain near the 1/2 position of the neutral axis of the test beam tended to zero. The strain of test beam L2 changed linearly along the height of the beam under load, and the strain of the Abaqus test beam model along the height of the beam conformed to the assumption of a plane section. The total strain of the test beam L2 under the ultimate load is shown in Figure 21.

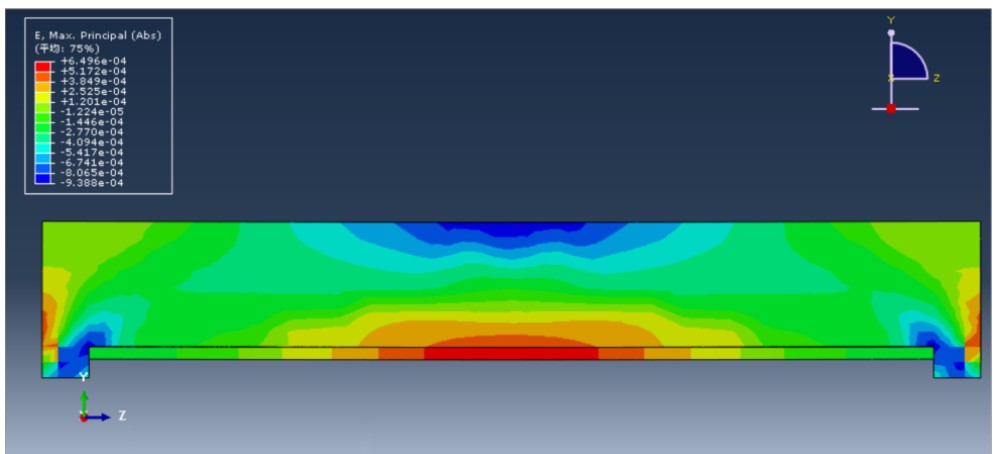

**Figure 21.** Total strain of test beam L2 under ultimate load.

**5. Conclusions**

By simulating the construction process in the reinforcement process of the real bridge, this test explored the test phenomenon, load–deflection, stiffness, ductility, and strain of test beam L2 under load after simulating the reinforcement of the real bridge, which was related to test beam L0 and test beam L1. For comparison, the flexural bearing capacity of reinforced test beam L2 was comprehensively evaluated. The following conclusions were drawn from the research:

(1)   Compared with test beam L1, the cracking load, yield load, and ultimate load of test beam L2 were reduced by 25%, 17.5%, and 15.4%, respectively. Compared with test beam L0, the cracking load, yield load, and ultimate load of test beam L2 were increased by 20%, 61.5%, and 62.5%, respectively. Test beam L2 significantly improved the crack resistance of the test beam by using prestressed steel strands and polyurethane toughening materials. The test beam effectively delayed the increasing trend of the deflection of the test beam after reinforcement, and the performance of delaying the deflection of the positive pouring method was slightly lower than that of test beam L1.

(2)   Compared with test beam L0, the stiffness of test beam L2 was increased by 85.3%, 47.9%, and 45.2%, respectively. Taking test beam L1 as a reference, the stiffness exhibited by test beam L2 decreased by 38.3%, 59.7%, and 39.1%, respectively. Although the forward pouring method was adopted, the combined toughening method of the prestressed steel strand and the polyurethane–cement composite material still played a role in enhancing the rigidity of the test beam, effectively improving the rigidity of the test beam.

(3)   Taking unreinforced beam L0 as a reference, the ductility of test beam L2 increased by 12.6%; compared with test beam L1, the ductility of test beam L2 decreased by 4.6%. The combined toughening of prestressed steel hinges and polyurethane–cement composites effectively improved the ground ductility of hollow slab beams and provided greater safety reserves for the subsequent application of real bridge reinforcement.

(4)   Combining the theoretical calculation value and the measured value of the bearing capacity, and the parameters such as deflection and strain, the flexural bearing capacity of the test beam was: test beam L1 > test beam L2 > test beam L0. The use of tensile prestressed steel strands and forward casting of polyurethane toughening materials effectively improved the flexural bearing capacity of the test beams, and this reinforcement process can be further extended to engineering applications.

(5)   Abaqus was used to simulate the test beam under load, and the simulated calculated value was in high agreement with the measured value, which verified the test phenomenon and the reliability of the test data of the three test beams under load.

**Author Contributions:** Conceptualization, T.Z. and X.L.; Methodology, D.X.; Software, T.Z.; Validation, D.X., D.C. and Z.L.; Formal analysis, J.L.; Investigation, J.L.; Resources, J.L.; Data curation, G.L.; Writing—original draft preparation, T.Z.; Writing—review and editing, J.L.; Visualization, X.L.; Supervision, T.Z.; Project administration, D.C.; Funding acquisition, D.X. All authors have read and agreed to the published version of the manuscript.

**Funding:** This study was supported by the Science and Technology Program of the Shandong Provincial Department of Transportation [No. 2017B97] and the Key R&D Program of Science and Technology Department of Shandong Province [No. 2019GGX102041].

**Institutional Review Board Statement:** Not applicable.

**Informed Consent Statement:** Not applicable.

**Data Availability Statement:** Not applicable.

**Conflicts of Interest:** The authors declare no conflict of interest.

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
