# Peer review of "Research on Flexural Bearing Capacity of Reinforced Hollow Slab Beams Based on Polyurethane Composite Material Positive and Negative Pouring Method"

_sustainability, doi:10.3390/su142417030_

Round 1
Reviewer 1 Report
This manuscript by Li et at demonstrated the great performance of prestressed steel strand-polyurethane cement composites in strengthening hollow slab beams by showing an inhibited development of concrete cracks and delayed damage process. Moreover, the findings reported here could be readily applied to other related areas, thus, this manuscript is recommended to publish on Sustainability after minor revisions as shown here:
1. The authors should clearly emphasize the importance of this study and how it could advance related fields.
2. Could the authors add standard deviations for the test parameters in Table 5 and 6 if applicable?
3. The authors were suggested to provide some information about the employed polyurethane, such as the possible molecular weight/structure/components, and some clues about how it reinforces the studied materials.
Author Response
- The importance of this research and how it advances related fields has been supplemented in the text.
- Standard deviations have been added in the text for the test parameters in Tables 5 and 6.
- The information on the composition of polyurethane has been supplemented in the text

Reviewer 2 Report
I find the work interesting and worthy of reporting. However, I suggest the authors to address the points bellow and submit a revised manuscript.
1) It would be very important to present SEM micrographs of cement composite before and after modification. Also, Energy Dispersive X-Ray Analysis (EDX) must be able to clarify how chemical compounds are distributed along the composite;
2) Also, SEM micrography at the steel/polyurethane interface would help author's claim that "the prestressed steel strand-polyurethane composite material is well bonded to the hollow slab beam";
3) The "positive and negative pouring method" is only mentioned in the Title and in the Conclusions. Perhaps the authors could extend the introduction section in order to contextualize this method for the journal's readership.
Author Response
- SEM micrographs have been added in the manuscript. Since the manuscript is more about the analysis of the macroscopic mechanical performance of the reinforced hollow slab beam, the EDX is not elaborated
- The strength and stiffness of the bonding section between the steel strand and polyurethane are very high, and sampling preparation is difficult, so the SEM electron microscope analysis of the interface was not carried out
- The comparison of positive and negative pouring methods has been explained in Table 4

Reviewer 3 Report
In this work, the polyurethane composite material was used for reinforcing hollow slab beams. The research is interesting, however, there are some questions to be solved before publication:
(1) How about the compatibility of cement and organic components?
(2) How to guarantee the synergistic curing of cement and polyurethane?
(3) Types of polyurethane will significantly affect the properties of composite materials, please illustrate the details of compositions for polyurethane.
(4) The performance of the cement-polyurethane composite material should be compared to previously reported results.
(5) There are a few related works available for the authors to refer to (Constr. Build Mater. 328 (2022) 126999, Constr. Build Mater. 138 (2017) 240, Mater. Design 117 (2017) 1).
Author Response
1, 2 The compatibility and synergy between cement and polyurethane have been supplemented in the text.
3 Information about the composition of polyurethane has been supplemented in the text.
4 The performance of cement-polyurethane composites has been compared with previously reported results in the paper
5 Relevant references have been added to the text

Round 2
Reviewer 2 Report
Most of the reviewers' suggestions/criticisms were addressed in the revised manuscript. The paper is now improved. Thus, I recommend the paper to be published as it is.
Reviewer 3 Report
In the revision, the authors have made proper changes to the manuscript and addressed all my concerns. I think the paper should be eligible for publication now.